# Efficacy and Safety of Xylitol Nasal Irrigation after Functional Endoscopic Sinus Surgery: A Randomized Controlled Study

**DOI:** 10.3390/biomedicines12061377

**Published:** 2024-06-20

**Authors:** Rong-San Jiang, Yi-Fang Chiang, Kai-Li Liang

**Affiliations:** 1Departments of Medical Research, Taichung Veterans General Hospital, Taichung 40705, Taiwan; rsjiang@vghtc.gov.tw; 2Departments of Otolaryngology, Taichung Veterans General Hospital, Taichung 40705, Taiwan; 3School of Medicine, Chung Shan Medical University, Taichung 40201, Taiwan; 4School of Medicine, College of Medicine, National Yang Ming Chiao Tung University, Taipei 112304, Taiwan; yvonnechiang9009@gmail.com; 5Department of Post-Baccalaureate Medicine, College of Medicine, National Chung-Hsing University, Taichung 40227, Taiwan

**Keywords:** chronic rhinosinusitis, functional endoscopic sinus surgery, nasal irrigation, normal saline, xylitol

## Abstract

Xylitol is considered a naturally occurring antibacterial agent. It is generally believed to enhance the body’s own innate bactericidal mechanisms. It also provides anti-adhesive effects against both Streptococcus pneumoniae and Haemophilus influenza. This study was performed to evaluate the efficacy and safety of xylitol nasal irrigation in the postoperative care of functional endoscopic sinus surgery (FESS). Patients with chronic rhinosinusitis who received FESS were recruited and randomly assigned to two groups at one month post-surgery. Thirty-five patients in the xylitol group received 400 mL of 5% xylitol nasal irrigation daily for 2 months, while another 35 in the normal saline (NS) group received 400 mL of NS nasal irrigation daily for 2 months. Prior to FESS, as well as before and after nasal irrigation, sinonasal symptoms were assessed through the 22-item Sino-Nasal Outcome Test Questionnaire. The patients also underwent an endoscopic examination while undergoing nasal function tests, and a cytokine measurement of the nasal lavage and a bacterial culture from the middle meatus were performed. The safety of the nasal irrigation was assessed through any self-reported adverse events, the Eustachian Tube Dysfunction Patient Questionnaire and the eustachian tube function test. The endoscopic scores and olfactory threshold significantly decreased after xylitol irrigation when compared with those before irrigation. The prevalence of Staphylococcus aureus in the nasal secretions also decreased significantly after xylitol irrigation. The amounts of Interleukin-5 and Interleukin-17A were significantly increased in the nasal lavage after xylitol irrigation. No side effects, including those related to eustachian tube function, were seen after nasal irrigation in both groups. Our results showed that xylitol nasal irrigation was both beneficial and safe during the postoperative care of FESS.

## 1. Introduction

Chronic rhinosinusitis (CRS) is a chronic inflammation of the nose and paranasal sinuses affecting approximately 14% of people in the United States [1]. CRS is considered to be a multifactorial disease. Many factors are related to its etiopathogenesis, including bacterial and fungal infection or colonization, allergies, anatomic anomalies and genetic predisposition [2,3]. Patients who do not respond to maximal medical treatment often need to undergo functional endoscopic sinus surgery (FESS) in order to relieve symptoms [4]. Postoperative care is essential in order to achieve the best surgical results [5]. During the postoperative period after FESS, nasal irrigation has been demonstrated to be an important method for promoting wound recovery in the nasal cavity and sinuses and as a method to reduce the use of unnecessary drugs [6]. It can improve mucociliary clearance, decrease edema, reduce the concentration of inflammatory mediators and reduce the amount of mucus accumulated in the cavities, in addition to preventing the formation of crusts [7]. When saline irrigation is widely used, anti-microbial agents, such as antibiotics or antifungal agents, may be added to the rinsing fluid [8,9].

Xylitol is a five-carbon sugar alcohol that is considered a naturally occurring antibacterial agent [10]. It is generally believed to enhance the body’s own innate bactericidal mechanisms [11]. It also provides anti-adhesive effects against both Streptococcus pneumoniae and Haemophilus influenza [12]. Xylitol use has several therapeutic purposes and the most established ones are the prevention of cavities and stabilization of glycemic levels, as it does not depend on insulin to be metabolized [7]. A few studies have demonstrated a significant sinonasal symptomatic benefit when using xylitol irrigation in CRS patients [7,10,11]. Xylitol nasal irrigation also results in a greater enhancement of nasal nitric oxide and inducible nitric oxide synthase mRNA in the maxillary sinus [13].

Xylitol is nontoxic and safe according to the US Food and Drug Administration, is absorbed slowly by the stomach and may cause loose stools when ingested in large amounts [7]. Oral xylitol is well tolerated in both adults and children [12]. One study has been performed to investigate the safety of inhaled xylitol, with the results showing that the aerosolization of iso-osmotic xylitol was both safe and well tolerated by human subjects [14]. Additionally, no change in spirometry was seen in the study subjects. No adverse effects have been reported with xylitol nasal irrigation, although the methods to evaluate the adverse effects have not been described in any studies [13,15]. The purpose of our study was to evaluate the efficacy and safety of xylitol nasal irrigation as a form of adjuvant therapy after FESS, in particular, the influence of nasal irrigation on eustachian tube function.

## 2. Materials and Methods

### 2.1. Study Population

CRS patients who had previously failed medical treatment and subsequently underwent bilateral primary FESS were enrolled in this study. The inclusion criteria were as follows: the diagnosis of CRS was made according to the EPOS criteria, based upon patient history, nasal endoscopy and a CT of the sinuses [16]. The exclusion criteria were as follows: we excluded patients with a history of immunodeficiency or sinus surgery, as well as those who had received antibiotic treatment within a week before FESS. Patients with a pathological diagnosis of fungal sinusitis or sinonasal tumor were also excluded. This study was approved by the Institutional Review Board (Ⅰ) of Taichung Veterans General Hospital (IRB No. CF19287A). Written consent was obtained from each patient. This clinical trial was registered at Clinicaltrial.gov (registration identifier: NCT06108921).

### 2.2. Nasal Irrigation

One month after undergoing FESS, 70 eligible patients were randomly assigned to the xylitol and the saline groups. Randomized assignment to the two groups was performed by an independent statistician, with the study investigators and patients being blinded to the group allocations and treatment modalities. Nasal irrigation was performed through use of a Sanvic SH903 pulsatile irrigator (Yun-Wang Industrial Co., Tainan, Taiwan). In the xylitol group, 5% xylitol solutions were first prepared by mixing two packets of 10 mg xylitol powder with 400 mL of sterile water in the container of the irrigator (Figure 1). In the saline group, the normal saline solution was prepared by mixing 2 packets of 1.8 mg salt powder with 400 mL of sterile water in the container of the irrigator (Figure 1). Patients were instructed to prepare the solutions by themselves at home. When irrigating the nose, patients irrigated each of their nasal cavities with 200 mL of solutions once a day. Patients in both groups performed nasal irrigation for 8 weeks, during which time, antibiotics, intranasal or oral antihistamines, and intranasal or oral steroids were not prescribed. No debridement was performed either.

### 2.3. Assessment of Efficacy

Prior to FESS, each patient completed a questionnaire, which was the Taiwanese version of the 22-item Sino-Nasal Outcome Test (TWSNOT-22) [17]. They also received an endoscopic examination, acoustic rhinometry, smell tests, saccharine transit test, and bacterial culture taken from the middle meatus.

Endoscopic appearances were quantified on a point scale of 0–2 according to the staging system designed by Lund and Mackay [18]. The endoscopic appearances were then categorized into polyps (0: no polyps; 1: polyps present within the middle meatus; 2: polyps beyond the middle meatus); nasal secretion (0: no secretion; 1: clear, thin secretion; 2: thick, purulent secretion); and mucosal edema; scarring; crusting; (0: absent; 1: mild; 2: severe). The scores ranged from 0 to 20 for both nostrils combined. The second minimal cross-sectional area (MCA_2_) of the nasal cavity was then measured by acoustic rhinometry. The MCA_2_ of both nostrils was averaged to give a mean MCA_2_ (cm^2^). The smell function was evaluated using the phenyl ethyl alcohol (PEA) threshold test [19] and the traditional Chinese version of the University of Pennsylvania Smell Identification Test (UPSIT-TC) (Sensonics, Inc., Hadden Heights, NJ, USA) [20]. The saccharine transit test involves placing saccharine granules under the head of the inferior turbinate in the more severely affected nostril. The time interval (minutes) between placement of saccharine granules and sensation of sweetness in the throat was recorded. Bacterial cultures were taken using a cotton-tipped stick which was put into both middle meatuses to collect swab samples under anterior rhinoscopy. The cotton tip was then placed in Thanswab tubes each containing 5 mL of Amies charcoal medium for culturing aerobes and anaerobes. At the laboratory, the swab cotton tip in a Thanswab tube was brushed on an agar plate containing 5% sheep blood, eosin methylene blue, and chocolate. The plate was incubated in an incubator with 5% CO_2_ at 35 °C for 2 and 4 days. A Brucella anaerobic blood agar was also inoculated for anaerobes, and subsequently incubated in the Form anaerobic system for 2 and 4 days. The specimen was finally placed into a thioglycollate broth tube for enriching anaerobes, and incubated at 35 °C for 2 days. All isolates were routinely examined in the laboratory, including checking for aerobic and facultative bacteria and anaerobes.

After FESS, patients were followed up with at the outpatient clinic, where crust and discharge were removed from the nasal cavities. Antibiotics, intranasal or oral antihistamines, and intranasal or oral steroids were not prescribed. A month after surgery, patients completed the TWSNOT-22 questionnaires, and received an endoscopic examination, acoustic rhinometry, smell tests, saccharine transit test, and bacterial culture once again. Finally, the patients were asked to breathe deeply inward and hold their breath. The nostril where the saccharine transit test was performed was irrigated with 20 mL of sterile water using a syringe. The irrigated fluid was then forcefully exhaled into a sterile pan, with the fluid in the pan poured into a centrifuge tube and transferred to the laboratory. In the laboratory, under a laminar flow hood, the fluid in the centrifuge tube was mixed with an equal volume of diluted dithiothreitol (1.055 mg/mL) and vortexed for 30 s. The tube was then kept at room temperature for 15 min while the dithiothreitol broke apart the disulfide bonds to liquefy the mucus, before being centrifuged once again at 4000 rpm for 10 min. The supernatant was collected and assayed for the presence of cytokines following the manufacturer’s protocol. The concentrations of human Interleukin-6 (IL6, ng/mL), interferon-gamma (IFN-γ, pg/mL), Interleukin-5 (IL5, pg/mL) and Interleukin-17A (IL17A, pg/mL) were all assessed using enzyme-linked immunosorbent assay (ELISA) kits (IL-5 and IFN-γ, BD OptEIA kit; IL-6 and IL-17A, invitrogen).

After completion of their 2-month nasal irrigation treatment, patients filled out the TWSNOT-22 questionnaire a final time. Each patient also received a second endoscopic examination, acoustic rhinometry, smell tests, saccharine transit test, and bacterial culture. The concentrations of IL6, IFN-γ, IL5 and IL17A were all measured in the nasal lavage as well.

### 2.4. Assessment of Safety

Any adverse effects surrounding nasal irrigation were evaluated by asking the subjects whether they had experienced any adverse events related to nasal irrigation, as well as performing blood tests, including a complete blood count, liver function and renal function test, and measuring eustachian tube function using the Eustachian Tube Dysfunction Patient Questionnaire (ETDQ-7) and eustachian tube function test using a GSI TympStar Pro™ (GRASON-STADLER, Eden Prairie, MN, USA).

The ETDQ-7 is a valid and reliable questionnaire used to evaluate eustachian tube dysfunction (ETD) [21]. It consists of 7 questions with each question having a value from 1 to 7. A score of 1 indicates no problem at all, while a score of 7 implies a very severe problem. A patient’s total score will range from 7 to 49 [22]. A traditional Chinese version of the ETDQ-7 has been recently developed. In this version, the optimal cutoff point for the total item score when diagnosing ETD was determined to be 13.5, with the sensitivity and specificity of this traditional Chinese version of the ETDQ-7 being 100% and 99.9%, respectively [23]. In this study, we used the traditional Chinese version of the ETDQ-7 in order to evaluate ETD, where a total item score of 14 or more was considered to be the threshold at which a patient would suffer from ETD.

The GSI TympStar Pro™ (GRASON-STADLER, Eden Prairie, MN, USA) uses a nine-step inflation/deflation test to measure eustachian tube function [24]. Failure to alter pressure in the middle ear by at least 10 daPa when swallowing during any of the steps was considered ETD (tubal function labeled as ‘Poor’). If the equilibration was successful (observed pressure change ≥10 daPa) in all steps, eustachian tube function was considered ‘Good’.

### 2.5. Endpoints of the Study

The primary endpoints of this study were evaluation of rhinosinusitis severity by the TWSNOT-22 questionnaire, an endoscopic examination, acoustic rhinometry, smell tests, and saccharine transit test after nasal irrigation, and assessment of safety of nasal irrigation by ETDQ-7 and eustachian tube function test. The secondary endpoints were bacterial culture results after nasal irrigation and the change in the concentrations of IL6, IFN-γ, IL5 and IL17A in the nasal lavage.

### 2.6. Sample Size and Statistical Analysis

The sample size was calculated through the study design of the Mann–Whitney U test using the power analysis program G* Power 3 [25]. Significant differences were set at 8.9 according to the results of validation of the SNOT-22 [26]. With an alpha value of 0.05 and a power value of 0.8, the results of the calculation indicated that approximately 35 patients were required for each group.

All data are presented as mean ± standard deviation. The gender of patients, bacterial culture rates, number of patients with ETD defined by ETDQ-7, and number of eustachian tubes with poor function were all compared between the xylitol and saline groups using the Pearson chi-square test. The age of patients, TWSNOT-22 score, endoscopic score, mean MCA_2_, PEA threshold, UPSIT-TC score, saccharine transit time, and the change in TWSNOT-22 score, endoscopic score, mean MCA_2_, PEA threshold, UPSIT-TC score, and saccharine transit time between the periods prior to surgery and before nasal irrigation, as well as between the periods prior to nasal irrigation and after nasal irrigation were all compared between the 2 groups through the Mann–Whitney U test. Generalized estimating equations were also used to perform an intergroup comparison. The TWSNOT-22 scores, endoscopic score, mean MCA_2_, PEA threshold, UPSIT-TC score, saccharine transit time, concentrations of IL6, IFN-γ, IL5 and IL17A, and the ETDQ-7 score were all compared between the periods prior to surgery and before nasal irrigation, as well as between the periods before nasal irrigation and after nasal irrigation using the Wilcoxon signed rank test. The bacterial culture rates, number of patients with ETD defined by ETDQ-7, and number of eustachian tubes with poor function were compared between the periods prior to surgery and before nasal irrigation, as well as between the periods before nasal irrigation and after nasal irrigation through the Pearson chi-square test. All computations were performed using SPSS (version 22.0, SPSS, Inc., Chicago, IL, USA). Two-tailed *p*-values < 0.05 were considered statistically significant.

## 3. Results

### 3.1. Patients

The flow chart and design of the experiment are shown in Figure 2. A total of 70 patients were included in the final analysis, with 35 being in the xylitol group and 35 in the saline group between February 2020 and February 2023. Table 1 shows the sex and age of the patients in both groups. No significant differences were found between the two groups in terms of age and gender (*p* = 0.565, 1, respectively).

### 3.2. Rhinosinusitis Severity Prior to FESS and before Nasal Irrigation

Table 2 and Table 3 show the severity of rhinosinusitis prior to FESS, before nasal irrigation, and after nasal irrigation in the xylitol and saline groups. The TWSNOT-22 score was 38.7 ± 19.9 before FESS and 30.2 ± 18.4 one month after FESS in the xylitol group, and 42.9 ± 23.1 before FESS and 24.5 ± 18.4 one month after FESS in the saline group. The endoscopic score was 5.1 ± 1.9 before FESS and 4.3 ± 1.6 one month after FESS in the xylitol group, and 5.1 ± 2.1 before FESS and 4.7 ± 2.0 one month after FESS in the saline group. The mean MCA_2_ was 0.41 ± 0.21 before FESS and 0.46 ± 0.15 one month after FESS in the xylitol group, and 0.40 ± 0.20 before FESS and 0.50 ± 0.16 one month after FESS in the saline group. The PEA threshold was −3.55 ± 2.73 before FESS and −3.91 ± 2.31 one month after FESS in the xylitol group, and −4.08 ± 2.94 before FESS and −4.99 ± 2.74 one month after FESS in the saline group. The UPSIT-TC score was 19.2 ± 7.3 before FESS and 22.6 ± 6.4 one month after FESS in the xylitol group, and 21.6 ± 8.1 before FESS and 26.1 ± 6.1 one month after FESS in the saline group. The saccharine transit time was 13.5 ± 7.5 min before FESS and 14.6 ± 7.6 one month after FESS in the xylitol group, and 14.0 ± 7.5 before FESS and 16.5 ± 9.8 one month after FESS in the saline group. The bacterial culture rate was 12.9% before FESS and 62.9% one month after FESS in the xylitol group, and 27.1% before FESS and 70% one month after FESS in the saline group. There were no significant differences in all the assessments of rhinosinusitis severity between the two groups. One month after FESS, both the TWSNOT-22 and endoscopic scores decreased significantly for patients in the xylitol group; however, MCA_2_ and the UPSIT-TC score increased significantly (Table 2). In the saline group, the TWSNOT-22 score decreased significantly, but MCA_2_ and the UPSIT-TC score increased significantly (Table 3). One month after FESS, there were no significant differences in all the assessments of rhinosinusitis severity between the two groups, except for the UPSIT-TC score, which was significantly higher for the patients in the saline group than those in the xylitol group (*p* = 0.012).

### 3.3. Comparison of Rhinosinusitis Severity between the Period Prior to Nasal Irrigation and after Nasal Irrigation

After a 2-month nasal irrigation period, the TWSNOT-22 score was 30.2 ± 18.4 before nasal irrigation and 23.9 ± 18.5 after nasal irrigation in the xylitol group, and 24.5 ± 18.4 before nasal irrigation and 17.9 ± 15.8 after nasal irrigation in the saline group. The endoscopic score was 4.3 ± 1.6 before nasal irrigation and 3.6 ± 1.5 after nasal irrigation in the xylitol group, and 4.7 ± 2.0 before nasal irrigation and 4.1 ± 1.6 after nasal irrigation in the saline group. The mean MCA_2_ was 0.46 ± 0.15 before nasal irrigation and 0.43 ± 0.14 after nasal irrigation in the xylitol group, and 0.50 ± 0.16 before nasal irrigation and 0.48 ± 0.19 after nasal irrigation in the saline group. The PEA threshold was −3.91 ± 2.31 before nasal irrigation and −4.59 ± 2.68 after nasal irrigation in the xylitol group, and −4.99 ± 2.74 before nasal irrigation and −5.10 ± 2.97 after nasal irrigation in the saline group. The UPSIT-TC score was 22.6 ± 6.4 before nasal irrigation and 23.5 ± 6.5 after nasal irrigation in the xylitol group, and 26.1 ± 6.1 before nasal irrigation and 25.5 ± 5.7 after nasal irrigation in the saline group. The saccharine transit time was 14.6 ± 7.6 min before nasal irrigation and 14.7 ± 8.1 after nasal irrigation in the xylitol group, and 16.5 ± 9.8 before nasal irrigation and 14.2 ± 7.1 after nasal irrigation in the saline group. The bacterial culture rate was 62.9% before nasal irrigation and 45.7% after nasal irrigation in the xylitol group, and 70% before nasal irrigation and 47.1% after nasal irrigation in the saline group. The endoscopic score and PEA threshold decreased significantly for patients in the xylitol group, with the TWSNOT-22 score and bacterial culture rate decreasing as well, although insignificantly (Table 2). Moreover, the prevalence of Staphylococcus aureus in the nasal secretions decreased significantly after 2 months of xylitol irrigation from 32 isolates/51 total isolates to 16 isolates/41 total isolates (*p* = 0.04); however, the prevalence of anaerobes increased significantly from 4 isolates/51 total isolates to 15 isolates/41 total isolates (*p* = 0.002) (Table 4). In the saline group, the TWSNOT-22 score and bacterial culture rate both decreased significantly (Table 3). However, the prevalence of Staphylococcus aureus in the nasal secretions did not decrease significantly after 2 months of saline irrigation, going only from 34 isolates/62 total isolates to 24 isolates/34 total isolates (*p* = 0.197) (Table 4). After 2 months of nasal irrigation, there were no significant differences in all the assessments of rhinosinusitis severity between the two groups. Table 5 shows the intergroup comparison of the severity of rhinosinusitis.

The concentrations of IL6, IFN-γ, IL5 and IL17A in the nasal lavage were compared in the period prior to nasal irrigation and after nasal irrigation (Table 2 and Table 3). The concentration of IL6 significantly decreased after nasal irrigation in both the xylitol and saline groups, but the concentration of IFN-γ did not change in either group. The concentrations of both IL5 and IL17A significantly increased after xylitol nasal irrigation, but did not achieve the same result after saline nasal irrigation (Figure 3).

### 3.4. Assessment of Safety

No adverse events were reported after 2 months of nasal irrigation in both the xylitol and saline groups, and complete blood counts as well as liver and renal functions were shown to be not affected either.

The change in the ETDQ-7 score is shown in Table 4. Prior to FESS, there were 18 patients in the xylitol group and 9 patients in the saline group whose ETDQ-7 score was 14 or more. Among the total 70 CRS patients, 27 (38.6%) suffered from ETD, as determined by the questionnaire evaluation.

One month after FESS but prior to nasal irrigation, there were 10 patients in the xylitol group and 9 patients in the saline group whose ETDQ-7 score was 14 or more. In the xylitol group, 10 patients’ ETDQ-7 score dropped from 14 or more to l3 or less; however, 2 patients’ ETDQ-7 score rose from l3 or less to 14 or more. In the saline group, five patients’ ETDQ-7 score dropped from 14 or more to l3 or less, but five patients’ETDQ-7 score rose from l3 or less to 14 or more.

After 2 months of irrigation treatment, 10 patients in the xylitol group and 4 patients in the saline group had ETDQ-7 scores of 14 or more. In the xylitol group, 10 patients’ ETDQ-7 score decreased from 14 or more to l3 or less; however, 2 patients’ ETDQ-7 score increased from l3 or less to 14 or more. The ETD ratio of CRS patients in the xylitol group did not change significantly after 2 months of irrigation treatment (*p* = 1.0). In the saline group, seven patients’ ETDQ-7 score decreased from 14 or more to l3 or less, but two patients’ ETDQ-7 score increased from l3 or less to 14 or more. The ETD ratio of the CRS patients in the saline group did not change significantly after 2 months of irrigation treatment (*p* = 0.219).

The change in eustachian tubal function when evaluated using a GSI TympStar Pro™ is shown in Table 6. Prior to FESS, there were 52 eustachian tubes of 31 patients in the xylitol group and 51 eustachian tubes of 31 patients in the saline group whose functions were poor. Among the total 70 CRS patients, 105 (72.9%) eustachian tubes suffered from ETD, as determined by the nine-step inflation/deflation test.

One month after FESS but prior to nasal irrigation, there were 54 eustachian tubes in the xylitol group and 50 eustachian tubes in the saline group whose functions were poor. In the xylitol group, seven tubes’ function changed from poor to good, while nine tubes’ function changed from good to poor. In the saline group, six tubes’ function changed from poor to good, while five tubes’ function changed from good to poor.

After 2 months of irrigation, there were 50 eustachian tubes in the xylitol group and 55 eustachian tubes in the saline group whose functions were poor. In the xylitol group, nine tubes’ function changed from poor to good, but five tubes’ function changed from good to poor. The ETD ratio of the eustachian tubes in the xylitol group did not change significantly after 2 months of irrigation treatment (*p* = 0.562). In the saline group, 7 tubes’ function changed from poor to good, but 12 tubes’ function changed from good to poor. The ETD ratio of the eustachian tubes in the saline group did not change significantly after 2 months of irrigation treatment (*p* = 0.435).

## 4. Discussion

Xylitol nasal irrigation during the post-FESS period has been reported to have sinonasal symptomatic benefits [7,11]. Our results show that xylitol nasal irrigation decreased endoscopic scores and improved smell function. The treatment also decreased the SNOT-22 scores, although insignificantly. In contrast, the saline nasal irrigation only decreased the SNOT-22 scores. This indicates that xylitol nasal irrigation was helpful during the post-FESS care.

Xylitol is a naturally occurring antibacterial five-carbon sugar alcohol which can inhibit the growth of Streptococcus pneumoniae in the presence of glucose [10,12]. It has been found to possess anti-adhesive effects against both Streptococcus pneumoniae and Haemophilus influenzae [27], while also being effective in inhibiting the formation of bacterial biofilm in the oral cavity [7]. In one experiment, after xylitol was used to wash the nasal cavity for a period of 4 days, the cultures obtained by nasal swabs showed that the number of coagulase-negative Staphylococcus significantly decreased when compared with those patients whose nasal cavities were washed with saline [15]. In our study, the bacterial culture rate and the number of Staphylococcus aureus bacteria increased one month after FESS in both groups as compared with those before FESS. There were two possible reasons for these results. One was that we collected the culture specimens from the middle meatuses before surgery, but we collected the culture specimens from the ethmoid cavities after FESS. The other possible reason was that the surgical wound might be a better environment for bacterial growth since we did not prescribe antibiotics after FESS. However, we found that the number of Staphylococcus aureus bacteria significantly decreased after 2 months of xylitol nasal irrigation, but the number of anaerobes significantly increased. This indicates that xylitol nasal irrigation possesses the ability to inhibit the growth of Staphylococcus aureus in the postoperative sinus cavity. Staphylococcus aureus has been suggested to play a dominant role in negatively affecting outcomes in FESS patients experiencing persistent postoperative symptoms, ongoing mucosal inflammation, and infections [28,29]. However, the number of anaerobes significantly increased after 2 months of xylitol nasal irrigation treatment. It has been proposed that increased relative abundances of anaerobic bacteria may influence Staphylococcus aureus physiology and pathogenesis [30].

In a systematic review of protein biomarkers in adult CRS patients, 14 biomarkers have been identified in nasal lavage fluid, including IL6, IFN-γ and IL5 [31]. IL6, which is secreted by macrophages in response to specific microbial molecules and is a member of the pro-inflammatory cytokine family, induces the expression of a variety of proteins responsible for acute inflammation [32]. Our results showed that the concentration of IL6 significantly decreased after nasal irrigation in both the xylitol and saline groups, which may have been caused by the decreased inflammation of the sinonasal mucosa after nasal irrigation treatment. CRS has been endotyped based on three major types of cell-mediated effector immunity: types 1, 2 and 3 (T1, T2 and T3, respectively) [33]. The T1 endotype is categorized by an elevated expression of the T1 cytokine IFN-γ; the T2 endotype is categorized by elevated T2 cytokines (IL4, IL5 and IL13) as well as measures of eosinophilic cation protein (ECP); and the T3 endotype is categorized by elevated T3 cytokines (IL17A and IL17F) [34]. In a study measuring cytokines in mucus taken from the middle meatus of post-FESS CRS patients, the concentrations of IFN-γ, ECP and IL 17A were all elevated in patients with postoperative tissue edema or nasal discharge when observed by nasal endoscopy when compared with those having normal endoscopic findings [34,35]. In our study, the concentration of IFN-γ did not change after nasal irrigation in both groups. On the other hand, the concentrations of IL5 and IL 17A significantly increased after xylitol nasal irrigation but did not achieve the same result after saline irrigation. T2 immunity can provide protection against different types of noninfectious noxious environmental factors, while T3 immunity is devoted to offering protection against extracellular bacteria and fungi [36]. It seemed that xylitol nasal irrigation increased the T2 and T3 immunity of the post-FESS sinonasal mucosa. IL17A has been considered to enhance the production of nitric oxide (NO) [37]. Another study has shown that xylitol nasal irrigation led to a significant increase in nasal NO [13]. The epithelium located in the paranasal sinuses is the major source of NO. It has been suggested that xylitol nasal irrigation may promote the repair of epithelial cells in the paranasal sinuses of CRS patients [13].

The safety of xylitol nasal irrigation has been reported in two studies [13,15]. Although no adverse effects were found, the methods to evaluate the adverse effects were not described in the studies. In our study, the reporting of any adverse event relative to nasal irrigation was requested after 2 months of nasal irrigation treatment in both the xylitol and saline groups, with no adverse events being reported in either group. We also examined complete blood counts as well as liver and renal function, none of which were affected by 2 months of nasal irrigation.

The effect of nasal irrigation on eustachian tube function has not yet been evaluated comprehensively. In this study, the eustachian tube function was evaluated subjectively using the ETDG-7 questionnaire, and objectively using the nine-step inflation/deflation test. Using the ETDQ-7, the prevalence of ETD in CRS patients was 48.5% in a Western report, 30% in a Korean study, and 25% in a Taiwanese study [38,39,40]. In this study, the prevalence of ETD was 38.6%. Eustachian tube function has rarely been measured objectively. Using the suggested diagnostic criteria of the nine-step inflation/deflation test, the prevalence of poor eustachian tube function in our patients was much higher than 38.6% (Table 4). However, the subjective and objective tests showed that the eustachian tube function was not affected by 2 months of xylitol nasal irrigation.

Although our study showed that xylitol nasal irrigation was both beneficial and safe during postoperative care of FESS, there were several limitations in our study. Our results did not give direct evidence to prove the benefit of xylitol nasal irrigation over saline nasal irrigation. As mentioned above, our results showed the concentration of IFN-γ did not change after nasal irrigation in both groups, but the concentrations of IL5 and IL 17A significantly increased after xylitol nasal irrigation but did not achieve the same result after saline irrigation. In our study, we did not classify our patients based on the endotypes. Xylitol nasal irrigation might be more helpful in some CRS endotypes.

## 5. Conclusions

Our results show that xylitol nasal irrigation used during the post-FESS period could improve sinonasal symptoms. The bacteriological study found that the load of Staphylococcus aureus in the nasal cavity decreased after xylitol nasal irrigation, while the concentrations of both IL5 and IL 17A in the nasal lavage increased after xylitol nasal irrigation, although their effects were not clear. The safety assessment determined that xylitol nasal irrigation was safe and did not affect eustachian tube function. In conclusion, xylitol nasal irrigation was both beneficial and safe during postoperative care of FESS.

## Figures and Tables

**Figure 1 biomedicines-12-01377-f001:**
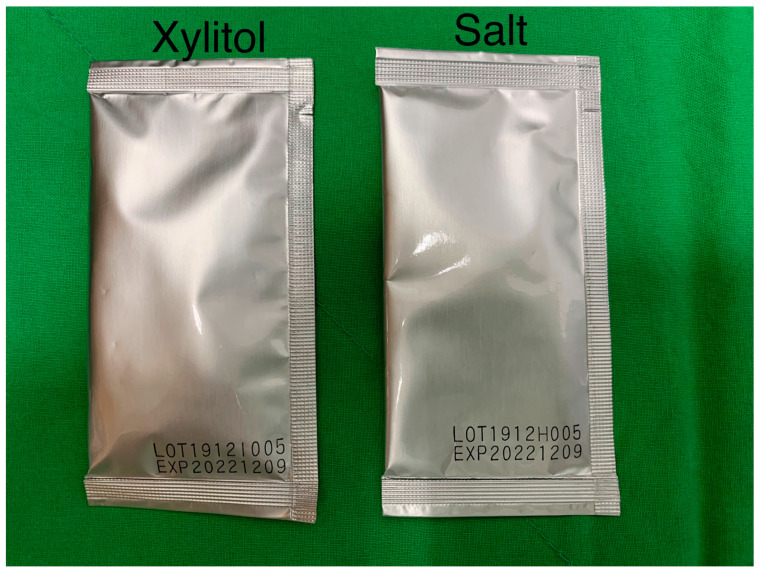
Packs of xylitol and salt powders.

**Figure 2 biomedicines-12-01377-f002:**
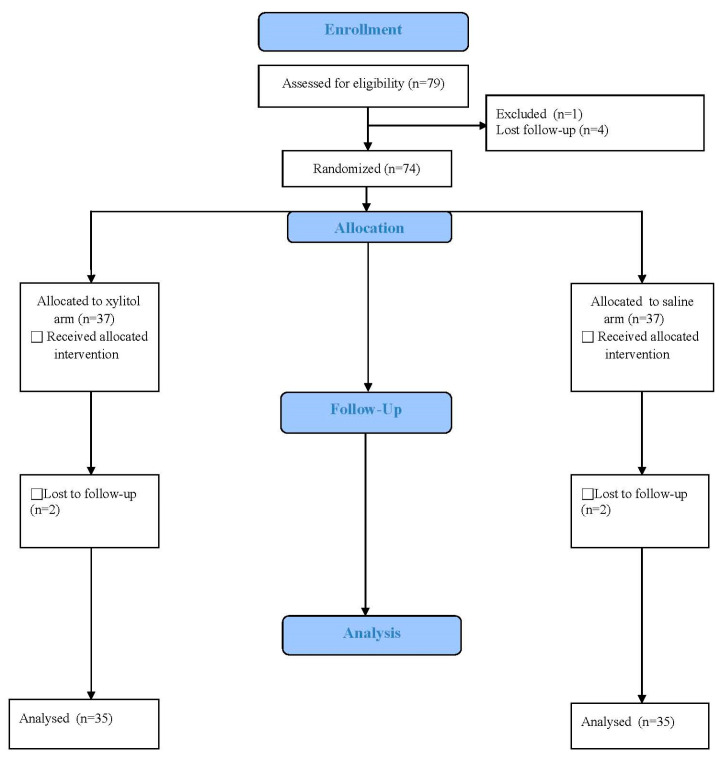
Flow chart showing steps from patient enrollment to data analyses.

**Figure 3 biomedicines-12-01377-f003:**
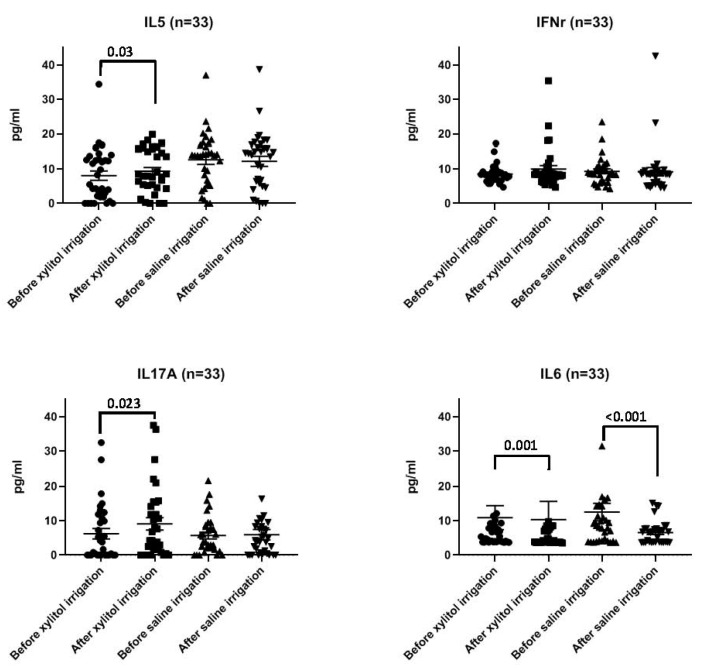
The concentrations of IL6, IFN-γ, IL5 and IL17A in the nasal lavage were compared at the periods both before nasal irrigation and after nasal irrigation.

**Table 1 biomedicines-12-01377-t001:** Demographics.

	Xylitol Group	Saline Group	*p* Value
Sex			
Male	20	21	1
Female	15	14	
Age	26–72 (50.1 ± 11.4) *	23–80 (47.9 ± 14.3)	0.565

*: range (mean).

**Table 2 biomedicines-12-01377-t002:** Rhinosinusitis severity in the xylitol group.

	Pre-OP	*p* Value *	Pre-NI	*p* Value **	Post-NI
TWSNOT-22 score	38.7 ± 19.9	0.014	30.2 ± 18.4	0.066	23.9 ± 18.5
Endoscopic score	5.1 ± 1.9	0.079	4.3 ± 1.6	0.022	3.6 ± 1.5
MCA_2_	0.41 ± 0.21	0.019	0.46 ± 0.15	0.064	0.43 ± 0.14
PEA threshold	−3.55 ± 2.73	0.708	−3.91 ± 2.31	0.042	−4.59 ± 2.68
UPSIT-C score	19.2 ± 7.3	0.028	22.6 ± 6.4	0.537	23.5 ± 6.5
Saccharine transit time	13.5 ± 7.5	0.384	14.6 ± 7.6	0.955	14.7 ± 8.1
Bacterial culture rate ***	9/70 (12.9%)	<0.001	44/70 (62.9%)	0.062	32/70 (45.7%)
IL6 concentration			10.875 ± 20.234	0.001	10.292 ± 30.376
IFN-γ concentration			8.519 ± 2.602	0.139	9.963 ± 5.927
IL5 concentration			8.064 ± 7.571	0.03	9.366 ± 6.139
IL17 concentration			6.457 ± 8.251	0.023	9.082 ± 10.253

Pre-OP: before surgery; Pre-NI: before nasal irrigation; Post-NI: after nasal irrigation; TWSNOT-22: 22-item; Sino-Nasal Outcome Test; MCA_2_: second minimal cross-sectional area; PEA: phenyl ethyl alcohol; UPSIT-C: traditional Chinese version of the University of Pennsylvania Smell Identification Test; *: comparison of rhinosinusitis severity in the periods both before FESS and before nasal irrigation; **: comparison of rhinosinusitis severity in the periods both before and after nasal irrigation; ***: positive culture sides/total culture sides of nasal cavities.

**Table 3 biomedicines-12-01377-t003:** Rhinosinusitis severity in the saline group.

	Pre-OP	*p* Value *	Pre-NI	*p* Value **	Post-NI
TWSNOT-22 score	42.9 ± 23.1	<0.0001	24.5 ± 18.4	0.001	17.9 ± 15.8
Endoscopic score	5.1 ± 2.1	0.304	4.7 ± 2.0	0.237	4.1 ± 1.6
MCA_2_	0.40 ± 0.20	<0.0001	0.50 ± 0.16	0.193	0.48 ± 0.19
PEA threshold	−4.08 ± 2.94	0.073	−4.99 ± 2.74	0.97	−5.10 ± 2.97
UPSIT-C score	21.6 ± 8.1	0.006	26.1 ± 6.1	0.482	25.5 ± 5.7
Saccharine transit time	14.0 ± 7.5	0.135	16.5 ± 9.8	0.362	14.2 ± 7.1
Bacterial culture rate ***	19/70 (27.1%)	<0.0001	49/70 (70%)	0.01	33/70 (47.1%)
IL6 concentration			12.523 ± 14.583	<0.001	10.292 ± 30.376
IFN-γ concentration			9.336 ± 3.894	0.145	9.210 ± 6.849
IL5 concentration			12.652 ± 7.775	0.164	12.213 ± 8.387
IL17 concentration			5.702 ± 5.628	0.721	5.933 ± 8.704

Pre-OP: before surgery; Pre-NI: before nasal irrigation; Post-NI: after nasal irrigation; TWSNOT-22: 22-item; Sino-Nasal Outcome Test; MCA_2_: second minimal cross-sectional area; PEA: phenyl ethyl alcohol; UPSIT-C: traditional Chinese version of the University of Pennsylvania Smell Identification Test; *: comparison of rhinosinusitis severity in the periods both before FESS and before nasal irrigation; **: comparison of rhinosinusitis severity in the periods both before and after nasal irrigation; ***: positive culture sides/total culture sides of nasal cavities.

**Table 4 biomedicines-12-01377-t004:** Bacteriology.

Group	Xylitol			Saline		
	Pre-OP	Pre-NI	Post-NI	Pre-OP	Pre-NI	Post-NI
Species			No. of Isolates			
Aerobic and facultative bacteria						
Gram-positive						
*Staphylococcus aureus*	4	32	16	6	34	24
Coagulase-negative staphylococci		3	3	3	2	
Staphylococcus not aureus			1	1		
*Corynebacterium* spp.					3	
*Streptococcus anginosus*		1				
Gram-negative						
*Citobacter koseri*	2	3	2	4	10	3
*Klebsiella pneumonia*		4			3	
*Klebsiella aerogenes*		1	1		1	3
*Haemophilus influenza*	1					
*Escherichia coli*					1	
*Pantoea* spp.					3	
*Pseudomonas aeruginosa*		2	1	1	3	1
*Acinetobacter junii*			1			
*Serratia marcescens*		1				
*Sphingomonas paucimobili*			1			
*Stenotrophomonas maltophilia*				1		
Total aerobic and facultative bacteria	7	47	26	16	60	31
Anaerobic bacteria						
Gram-positive						
*Propionibacterium acnes*	1	1	5	3	1	2
*Propionibacterium granulosum*			1			
*Propionibacterium avidum*			1			
*Slackia exigua*	1					
*Pavimonas micra*		1	1	1		1
*Finegoldia magna*			6			
Gram-negative						
*Fusobacterium nucleatum*	1	1				
*Veillonella atypica*				2	1	
*Bacteroides fragilis*		1	1			
*Bacteroides thetaiotaomicron*				1		
*Porphyromonas gingivalis*				1		
Total anaerobic bacteria	3	4	15	8	2	3
Total bacterial isolates	19	51	41	24	62	34

Pre-OP: before surgery; Pre-NI: before nasal irrigation; Post-NI: after nasal irrigation.

**Table 5 biomedicines-12-01377-t005:** Intergroup and intragroup comparison of the severity of rhinosinusitis.

	TWSNOT-22 Score				Endoscopic Score				MCA_2_			
	B	(95% CI)		*p* Value	B	(95% CI)		*p* Value	B	(95% CI)		*p* Value
Group												
Saline	ref.			ref.	ref.				ref.			
Xylitol	1.79	(−5.81–9.39)		1.79	−0.32	(−0.88–0.24)		0.263	−0.02	(−0.09–0.05)		0.557
Time												
Pre-OP	ref.			ref.	ref.				ref.			
Pre-NI	−13.47	(−18.08–8.87)		−13.47	−0.61	(−1.22–0.01)		0.047 *	0.08	(0.04–0.12)		<0.001 **
Post-NI	−19.90	(−24.28–15.52)		−19.90	−1.23	(−1.73–0.73)		<0.001 **	0.05	(0.01–0.10)		0.019 *
	PEA threshold				UPSIT-C score				Saccharine transit time			
	B	(95% CI)		*p* value	B	(95% CI)		*p* value	B	(95% CI)		*p* value
Group												
Saline	ref.				ref.				ref.			
Xylitol	0.64	(−0.40–1.67)		0.227	−2.45	(−5.01–0.10)		0.060	−0.45	(−3.18–2.28)		0.747
Time												
Pre-OP	ref.				ref.				ref.			
Pre-NI	−0.63	(−1.34–0.07)		0.079	3.90	(2.04–5.76)		<0.001 **	1.79	(−0.50–4.07)		0.125
Post-NI	−1.03	(−1.73–0.33)		0.004 **	4.07	(2.40–5.75)		<0.001 **	0.69	(−1.31–2.68)		0.500

*: <0.05; **: <0.01.

**Table 6 biomedicines-12-01377-t006:** Change in eustachian tube function.

	Pre-OP	*p* Value *	Pre-NI	*p* Value **	Post-NI
Xylitol group (35 ^a^)					
ETDQ-7 score	15.3 ± 9.7	0.003	11.3 ± 4.9	0.909	11.9 ± 6.6
ETDQ-7 score ≥ 14	18 ^a^ (51.4%)	0.088	10 (28.6%)	1	10 (28.6%)
Poor tubal function ^b^	52 (74.3%)	0.844	54 (77.1%)	0.562	50 (71.4%)
Saline group (35 ^a^)					
ETDQ-7 score	13.9 ± 11.0	0.032	10.3 ± 5.1	0.072	9.0 ± 3.5
ETDQ-7 score ≥ 14	9 ^a^ (25.7%)	1	9 (25.7%)	0.219	4 (11.4%)
Poor tubal function ^b^	51 (72.9%)	1	50 (71.4%)	0.435	55 (78.6%)

Pre-OP: before surgery; Pre-NI: before nasal irrigation; Post-NI: after nasal irrigation; *: comparison of eustachian tube function in the periods both before FESS and before nasal irrigation; **: comparison of eustachian tube function in the periods both before and after nasal irrigation; ^a^: number of patients; ^b^: number of eustachian tubes experiencing poor function as determined by the nine-step inflation/deflation test.

## Data Availability

The data presented in this study are available on request from the corresponding author.

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
