# Peer review of "Efficacy and Safety of Xylitol Nasal Irrigation after Functional Endoscopic Sinus Surgery: A Randomized Controlled Study"

_biomedicines, 2024, doi:10.3390/biomedicines12061377_

Round 1

Reviewer 1 Report

Comments and Suggestions for Authors

comparisons are indirect and a benefit of xylitol over normal saline is not shown. Maybe an emphasis on the discussion on the differences between the two groups upon the use of the appropriate statistical test that allow comparissons of changes should be provided. 

Author Response

comparisons are indirect and a benefit of xylitol over normal saline is not shown. Maybe an emphasis on the discussion on the differences between the two groups upon the use of the appropriate statistical test that allow comparissons of changes should be provided. 

Answer: Thanks for your comments. Yes, our results did not give direct evidences to prove a benefit of xylitol nasal irrigation over saline nasal irrigation. We have addressed this as a limitation of our study. On the other hand, we used the Mann-Whitney U test to compare the change of TWSNOT-22 score, endoscopic score, mean MCA2, PEA threshold, UPSIT-TC score, and saccharine transit time between the periods prior to surgery and before nasal irrigation, as well as between the periods prior to nasal irrigation and after nasal irrigation between the 2 groups. The results were shown in Tables 4 and 6.

Reviewer 2 Report

Comments and Suggestions for Authors

Abstract:
Give a detailed explanation of the study's objectives, design, patient population, main outcomes, important efficacy findings, safety precautions, and conclusive remarks.

Introduction:
- Extend the supporting data for nasal irrigation following FESS

- introduce all the genetic variations predisposing CRS. discuss and cite     doi:10.1111/coa.13870 and doi:10.1016/j.otc.2016.08.009.
- Give further xylitol background information and previous efficacy data.
The purpose and justification for the current investigation

Methods:
- Define the inclusion and exclusion standards.
- Give specifics about the randomization procedure
- Indicate the main and secondary results.
- Provide information about statistical analysis and sample size calculation.

Results:
- Include a participant flow diagram
- Provide the baseline attributes table.
- Provide descriptive data both before and after the course of treatment for both groups.
Provide statistical tests along with between-group comparisons.
- Only present findings that are clinically or statistically significant.

Discussion:
- Condense important discoveries and analyze outcomes  . cite relevant     10.1007/s00405-023-08184-6 and PMID:33085349.
- Talk about bad incidents and safety.
- Address biases and limitations in the study
- Draw precise conclusions regarding safety and efficacy.

Comments on the Quality of English Language

any

Author Response

Abstract:
Give a detailed explanation of the study's objectives, design, patient population, main outcomes, important efficacy findings, safety precautions, and conclusive remarks.

Answer: Thanks for your comments. We have revised our abstract based on your comments.

Introduction:
- Extend the supporting data for nasal irrigation following FESS

Answer: Thanks for your comments. We have expanded the supporting data for nasal irrigation following FESS.

- introduce all the genetic variations predisposing CRS. discuss and cite     doi:10.1111/coa.13870 and doi:10.1016/j.otc.2016.08.009.

Answer: Thanks for your comments. We had added a paragraph to introduce the genetic variations predisposing CRS and cite doi:10.1111/coa.13870 and doi:10.1016/j.otc.2016.08.009 as references 2 and 3.

- Give further xylitol background information and previous efficacy data.

Answer: Thanks for your comments. We had added further xylitol background information and previous efficacy data.

The purpose and justification for the current investigation
Answer: Thanks for your comments. The purpose and justification for the current investigation have been provided.

Methods:
- Define the inclusion and exclusion standards.

Answer: Thanks for your comments. We have defined the inclusion and exclusion criteria.

- Give specifics about the randomization procedure

Answer: Thanks for your comments. The randomization procedure was as follows.  Randomized assignment to the two groups was performed by an independent statistician, with the study investigators and patients being blinded to the group allocations and treatment modalities.

- Indicate the main and secondary results.

Answer: Thanks for your comments. The primary endpoints are evaluation of rhinosinusitis severity by TWSNOT-22 questionnaire, a endoscopic examination, acoustic rhinometry, smell tests, and saccharine transit test after nasal irrigation, and assessment of safety of nasal irrigation by ETDQ-7 and Eustachian tube function test. The secondary endpoints were bacterial culture results after nasal irrigation and the change of the concentrations of IL6, IFN-γ, IL5 and IL17A in the nasal lavage.

- Provide information about statistical analysis and sample size calculation.
Answer: Thanks for your comments. The sample size was calculated through the study design of the Mann-Whitney U test using the power analysis program G* Power 3. Significant differences were set at 8.9 according to the results of validation of the SNOT-22. With an alpha value of 0.05 and a power value of 0.8, the results of the calculation indicated that approximately 35 patients were required for each group.

Results:
- Include a participant flow diagram

Answer: Thanks for your comments. The flow chart and design of the experiment are shown in Figure 2.

- Provide the baseline attributes table.

Answer: Thanks for your comments. The demographics were shown in Table 1.

- Provide descriptive data both before and after the course of treatment for both groups.

Answer: Thanks for your comments. We have provided descriptive date both before and after the course of treatment for both groups in the Results section.

Provide statistical tests along with between-group comparisons.

Answer: Thanks for your comments. We used the Mann-Whitney U test to compare the change of TWSNOT-22 score, endoscopic score, mean MCA2, PEA threshold, UPSIT-TC score, and saccharine transit time between the periods prior to surgery and before nasal irrigation, as well as between the periods prior to nasal irrigation and after nasal irrigation between the 2 groups. The results were shown in Tables 4 and 6.

- Only present findings that are clinically or statistically significant.

Answer: Thanks for your comments. We have removed the statistically insignificant findings.

Discussion:
- Condense important discoveries and analyze outcomes  . cite relevant     10.1007/s00405-023-08184-6 and PMID:33085349.

Answer: Thanks for your comments. We have condense our important discoveries and cite 10.1007/s00405-023-08184-6 and PMID:33085349 as references 33 and 35.

- Talk about bad incidents and safety.

Answer: Thanks for your comments. We have reported bad incidents and safety of nasal irrigation using self-reported adverse event, blood test and E-tube function test

- Address biases and limitations in the study

Answer: Thanks for your comments. We added a paragraph to address the limitations of our study.

- Draw precise conclusions regarding safety and efficacy.

Answer: Thanks for your comments. We added a statement to conclude the efficacy and safety of xylitol nasal irrigation.

Reviewer 3 Report

Comments and Suggestions for Authors

I would ask the authors to comment on one of their study findings that has not been done in the discussion section. How do they explain such significant increase of positive cultures 2 months after FESS, which number was surprisingly low before the surgery (12.9% in the xylitol group and and 27.1% in the saline group). Two months after surgery the number of positive cultures grew up to 62.9% and 70% respectively. Number of positive S. aureus cultures increased from 4 to 32 in the xylitol group, same thing happened in the saline group.

Bacterial culture technique should be detailed in the Material and Methods section.

Author Response

I would ask the authors to comment on one of their study findings that has not been done in the discussion section. How do they explain such significant increase of positive cultures 2 months after FESS, which number was surprisingly low before the surgery (12.9% in the xylitol group and 27.1% in the saline group). Two months after surgery the number of positive cultures grew up to 62.9% and 70% respectively. Number of positive S. aureus cultures increased from 4 to 32 in the xylitol group, same thing happened in the saline group.

Answer: Thanks for your comments.

We added a paragraph in the discussion section to explain why the bacterial culture rate and the number of S. aureus increased after FESS as compared with those before FESS. There were two possible reasons for these results. One was that we collected the culture specimens from the middl meati before surgery, but we collected the culture specimens from the ethmoid cavities after FESS. The other possible reason was that the surgical wound might be a better environment for bacterial growth since we did not prescribe antibiotics after FESS.

Bacterial culture technique should be detailed in the Material and Methods section.

Answer: Thanks for your comments.

We provided a detailed description of the bacterial culture technique in the Material and Methods section.

The cotton-tip was then placed in Thanswab tubes each containing 5 ml of Amies charcoal medium for culturing aerobes and anaerobes. At the laboratory, the swab cotton-tip in a Thanswab tube was brushed on an agar plate containing 5% sheep blood, eosin methylene blue, and chocolate. The plate was incubated in an incubator with5% CO2 at 35°C for 2 and 4 days. A Brucella anaerobic blood agar was also inoculated for anaerobes inoculated for anaerobes, and subsequently incubated in the Form anaerobic system for 2 and 4 days. The specimen was finally placed into a thioglycollate broth tube for enriching anaerobes, and incubated at 35 °C for 2 days. All isolates were routinely examined in the laboratory, including checking for aerobic and facultative bacteria and anaerobes.

Round 2

Reviewer 1 Report

Comments and Suggestions for Authors

My main initial concern (which remains) was that the authors randomized the patients to xylitol and normal saline (ns) groups but they compared the groups indirectly. For example they found that the change in endoscopic score was significant in xylitol group but not in ns group. Although their data are detailed enough and their conclusions seem valid, maybe they should consult a statistician to propose an appropriate test for this comparisons eg a type of kalpan- mayer?

In general their paper is interesting and provides a lot of useful data and hypothesis on the effect of xylitol in fess patients

Author Response

My main initial concern (which remains) was that the authors randomized the patients to xylitol and normal saline (ns) groups but they compared the groups indirectly. For example they found that the change in endoscopic score was significant in xylitol group but not in ns group. Although their data are detailed enough and their conclusions seem valid, maybe they should consult a statistician to propose an appropriate test for this comparisons eg a type of kalpan- mayer?

In general their paper is interesting and provides a lot of useful data and hypothesis on the effect of xylitol in fess patients

Answer: Thanks for your comments. We have consulted a biostatistician and she suggested to use Generalized Estimating Equations to perform intergroup comparisons. We have changed our table.